# Silver Birch (*Betula pendula* Roth.) on Dry Mineral Rather than on Deep Peat Soils Is More Dependent on Frozen Conditions in Terms of Wind Damage in the Eastern Baltic Region

**DOI:** 10.3390/plants11091174

**Published:** 2022-04-26

**Authors:** Oskars Krišāns, Roberts Matisons, Jānis Vuguls, Steffen Rust, Didzis Elferts, Andris Seipulis, Renāte Saleniece, Āris Jansons

**Affiliations:** 1Latvian State Forest Research Institute ‘Silava’, 111 Rigas Street, LV-2169 Salaspils, Latvia; oskars.krisans@silava.lv (O.K.); roberts.matisons@silava.lv (R.M.); janis.vuguls@silava.lv (J.V.); didzis.elferts@lu.lv (D.E.); andris.seipulis@silava.lv (A.S.); renate.saleniece@silava.lv (R.S.); 2Faculty of Resource Management, University of Applied Sciences and Arts, Büsgenweg 1a, 37077 Göttingen, Germany; steffen.rust@hawk.de; 3Faculty of Biology, University of Latvia, 1 Jelgavas Street, LV-1004 Riga, Latvia

**Keywords:** static tree-pulling tests, wind resistance, wind damage, frozen soil, silver birch, *Betula pendula*

## Abstract

In Northern Europe, the ongoing winter warming along with increasing precipitation shortens the periods for which soil is frozen, which aggravates the susceptibility of forest stands to wind damage under an increasing frequency of severe wind events via the reduction in soil–root anchorage. Such processes are recognized to be explicit in moist and loose soils, such as deep peat, while stands on dry mineral soils are considered more stable. In the hemiboreal forest zone in the Eastern Baltics, silver birch (*Betula pendula* Roth.) is an economically important species widespread on mineral and peat soils. Although birch is considered to be less prone to wind loading during dormant periods, wind damage arises under moist and non-frozen soil conditions. Static tree-pulling tests were applied to compare the mechanical stability of silver birch on frozen and non-frozen freely draining mineral and drained deep peat soils. Basal bending moment, stem strength, and soil–root plate volume were used as stability proxies. Under frozen soil conditions, the mechanical stability of silver birch was substantially improved on both soils due to boosted soil–root anchorage and a concomitant increase in stem strength. However, a relative improvement in soil–root anchorage by frozen conditions was estimated on mineral soil, which might be attributed to root distribution. The soil–root plates on the mineral soil were narrower, providing lower leverage, and thus freezing conditions had a higher effect on stability. Accordingly, silver birch on peat soil had an overall higher estimated loading resistance, which suggested its suitability for forest regeneration on loose and moist soils within the Eastern Baltic region. Nevertheless, adaptive forest management supporting individual tree stability is still encouraged.

## 1. Introduction

In European forests, the frequency and magnitude of wind disturbances are steadily increasing, causing substantial socio-economic and ecological impacts [1,2,3,4]. In Northern Europe and particularly in the Eastern Baltic region, the strongest wind events causing most of the damage to standing stock occur during late autumn–early spring [5,6]. During that period, the susceptibility of forest stands to wind damage is notably decreased by the freezing of soil [7,8]. However, the ongoing warming in synergy with the increasing heterogeneity in precipitation (deeper snow cover) during the dormancy period are shortening the period for which soil is frozen [9,10]. Furthermore, such conditions reinforced by increased soil moisture saturation reduce the stability of trees, making them susceptible to less severe wind events [11,12].

Tree mechanical stability primarily depends on the strength of soil–root anchorage and stem strength, and deficiency in either of them leads to tree failure through processes such as uprooting or stem breakage [8,13]. Regarding the uprooting, frozen soil improves soil–root anchorage through the reinforcement of binding between soil particles [14], thus reducing the risk of failure. At this point, increased soil stiffness reduces root movement, which enhances tree resistance against uprooting [14]. The sufficient soil–root anchorage predetermines tree mechanical stability to be dependent on stem strength [13], which is also reported to increase under the frozen conditions [8,15]. However, tree failure occurs in two stages [16,17]. Prior to the fatal (secondary) failure at the maximum loading (SF), internal primary wood failure (PF), which is not apparent during the stem bending, occurs [16,17]. This PF alters tree hydraulics, which can cause physiological drought stress [16,17], subjecting the storm-surviving trees to legacy effects [18,19]. Therefore, the shortening of the soil freezing period might also contribute to increasing the frequency of synergic legacy effects of wind events [20,21], including on wind-tolerant species [22]. 

Birch (*Betula* spp.) is a widespread tree species with high (and growing) economic and ecological importance in Northern Europe, particularly in the Eastern Baltic region [23]. It can successfully regenerate in various types of growing conditions [24]. The species is considered to be wind-tolerant, particularly during the dormant (leafless) period [25,26]. In the Eastern Baltic region, most birch stands are situated on freely draining mineral soils, yet a considerable share of the standing stock grows on peat soils [27,28] and is productive there. However, stands on peat soils are reported to be more prone to wind damage [5] due to the lower soil bearing capacity and shallow water table reducing rooting depth [29,30,31]. Furthermore, stands on peat soils are considered to be particularly prone to wind damage during the dormant period due to the shorter period of frozen soil and shallower freezing depth [5,7].

As species with an extensive range, there are indications of local adaptations of metapopulations of birch to wind conditions [32], as highlighted by local and regional differences in frequency of wind damage [5,25,26]. Accordingly, the aim of this study was to assess the increase in the loading resistance of birch under frozen soil conditions on mineral and peat soils in the Eastern Baltic region in Latvia. We hypothesize that soil freezing has a greater effect on the loading resistance of birch growing on peat soils. Information about tree mechanical stability can be obtained by the static tree-pulling test, which is a method used for the assessment of the strength of stem and soil–root anchorage [8,13,32,33]. This approach can also be successfully utilized to evaluate the differences in tree loading resistance between frozen and non-frozen soils [8]. 

## 2. Results

In the Eastern Baltic region, frozen soil conditions improved the soil–root anchorage of the studied mid-aged silver birch stands growing on freely draining mineral and drained peat soils. Frozen soil depth was higher in mineral than in peat soil, reaching 34 ± 1 cm and 20 ± 2 cm, respectively; however, the depth of frozen peat soil was apparently limited by the level of the groundwater table. As a result, the mechanical stability of silver birch became more stem strength-dependent, as nearly all sample trees experienced stem breakage under frozen soil conditions. Soil–root anchorage was particularly enhanced by the freezing of mineral soil, as the frequency of stem breakage reached 88% vs. 5% (single tree) under non-frozen conditions. For peat soil, the frequency of stem breakage increased less, from 69% to 80%. However, stem flexibility, as indicated by the modulus of elasticity (MOE), was affected by neither freezing conditions nor soil type, and stem resistance against breakage was improved, as indicated by the significant (*p* < 0.01, χ^2^ = 6.82) increase in the modulus of rupture (MOR) (Figure 1, Table 1). However, this comparison was possible for peat soil only, as an insufficient (in terms of statistical analysis) number of trees broke on non-frozen mineral soil. 

As a result of improved soil–root anchorage and stem strength, trees were estimated to have considerably higher static loading resistance under frozen soil conditions, as indicated by the significant increase in basal bending moments both at PF (*p* = 0.04, χ^2^ = 3.85) and particularly at SF (*p* < 0.001, χ^2^ = 45.61) (BBM_PF_ and BBM_SF_, respectively; Figure 1, Table 1). The relative increase in BBM_PF_ was 17% and 12% for peat and mineral soil, respectively. The increase (relative) in BBM_SF_ was more explicit, as the maximum loading resistance increased by 28% on peat soil and by 38% on mineral soil (Figure 1, Table 1). Furthermore, the maximum loading resistance differed significantly between soil types, though the effect was weak (*p* = 0.04, χ^2^ = 4.00).

Surprisingly, under non-frozen conditions BBM_SF_ was relatively lower on mineral soil by 19%; however, this difference was reduced to 6% under frozen soil conditions, indicating a stronger effect for birch on freely draining mineral soil. Such differences might be related to the volume of the root-soil plate, which is a direct proxy of mechanical stability. The soil–root plate was estimated to be wider and hence larger on peat soil (*p* < 0.001, χ^2^ = 11.10); however, it did not differ according to the soil freezing conditions (Figure 1, Table 1), supporting the importance of soil stiffness in the mechanical stability of silver birch. The estimated effects of soil freezing conditions and soil type were little affected by local adaptation, as indicated by the low influence of stand represented by the intraclass correlation coefficient (ICC ≤ 0.13; Table 1). Furthermore, data on tree size, soil type, and soil freezing conditions were sufficient for the estimation of BBM_SF_, MOR, and the volume of the soil–root plate, as indicated by R^2^ ≥ 0.37 for the relevant statistical model. 

## 3. Discussion

Tree survival during severe wind events depends on capability to resist uprooting or stem breakage [34]. Frozen soil conditions had a systematic positive impact on the mechanical stability of silver birch, as indicated by non-interacted effect, improving the static loading resistance of trees substantially. Locality (site) had little effect on the proxies of mechanical stability, as indicated by the low variance related to stand. Accordingly, the risk of wind damage is reduced under frozen soil conditions, which is in accordance with earlier studies [8,15]. However, the effect of frozen soil conditions on static loading resistance differed between soil types.

The relative improvement in soil–root anchorage by frozen conditions on mineral soil, as indicated by the stronger increase in the frequency of stem breakage and given that the volume of soil–root plate remained the same (Figure 1, Table 1), was unexpected. The increase in soil–root anchorage under the frozen conditions appears through the reinforcement of the binding between soil particles [14], as freezing increases soil stiffness. Therefore, the better improvement of soil–root anchorage on mineral soil can be explained by the deeper levels of soil freezing, which might be related to the higher thermal conductivity [35] and deeper groundwater table compared to deep peat [36]. 

Stands on peat soil, however, were estimated to have higher potential to resist fatal failure, as indicated by higher BBM_SF_ (Figure 1, Table 1), although the relative improvement of soil–root anchorage was less pronounced compared with that of mineral soil. This can be explained by differences in root distribution, as on peat soil the soil–root plates were wider, providing higher leverage [37]. Furthermore, trees on less stable soils increase the stiffness of their structural roots to cope with soil damping effects [38,39]. The low frequency of uprooted trees on peat soils contradicted the common assumption that uprooting resistance depends on rooting depth more than on the strength of lateral roots [29,40], thus supporting the stability of trees on deep peat soils [40].

Under improved soil–root anchorage (Figure 1, Table 1), tree mechanical stability became more stem strength-dependent [8,13], as the resistance against fatal failure increased. As expected, the maximum stem strength increased with freezing conditions [8,15], which was indicated by a higher MOR, thus implying a higher loading resistance against stem breakage. Nevertheless, we speculate that the increase in stem resistance might be a result of the underestimation of soil damping, as sufficient soil stiffness prevented the rotational movement of the soil–root plate under the static loading [33], unlike in non-frozen conditions. As a result, stems might be subjected to extra loading, which otherwise would be absorbed by the soil–root plate.

Enhanced stem strength could also be attributed to the marginal increase in BBM_PF_ (Figure 1, Table 1), suggesting that frozen conditions might reduce wood fiber kinking [16,17]. However, even apparently slight effects might have far-reaching consequences [21,41], as PF implies irreversible internal wood damage [16,17,42], which might notably affect stem hydraulic architecture [16,17]. Such damage can cause negative legacy effects, as a relatively long period of time might be needed for trees to recover from PF [42]. Therefore, the re-occurrence and accumulation of structural damage can further increase the risk of tree weakening and the ongoing reduction in loading resistance. 

Soil–root anchorage is considered to be largely dependent on microsite conditions [43], which determine the distribution and connectivity of roots and soil–root interactions [32,44]. However, frozen soil did not influence such microsite conditions, as indicated by the comparable confidence intervals of tested stability proxies (Figure 1). Microsite conditions can vary widely within a stand, particularly in naturally regenerated ones, thus resulting in heterogeneity in tree physical properties [45], which was apparently stronger for the elastic deformation of the stem, as shown by MOE (Figure 1). The lack of a significant effect of freezing conditions on MOE was surprising, as a positive effect has been observed for birch before [46]. However, considering the similar conditions during the testing, the divergence from previous observations [46] appear reasonable, suggesting the effect of local/regional adaptations of trees to climatic conditions including wind [47,48].

The observed changes in soil–root anchorage and the physical properties of tree stems highlights the reduced tree mechanical stability under non-frozen soil conditions. A similar reaction of tested trees implies common characteristics in the adaptation of trees within a provenance region, thus suggesting wind tolerance to be an important trait for tree survival and reproduction [41]. Moreover, silver birch is reported to have an ability of a universal adaptation to wind loading regardless of the growing environment [13]. However, the reduction in soil–root anchorage due to non-frozen soil conditions is likely to increase the risk of wind damage, while soil freezing is still common. Therefore, the adaptation of soil–root anchorage to shifts in soil freezing might limit the distribution of less stable genotypes, as they are subjected to a higher risk of negative legacy effects following severe wind events. The reduction in tree mechanical stability under non-frozen soil conditions indicates the necessity of the adaptation of trees and silviculture practices in order to mitigate a potential negative effect of winter warming. Furthermore, the ongoing warming in synergy with the increase in precipitation reduce the soil–root anchorage, making trees even more prone to failure during severe wind events [11,12].

## 4. Materials and Methods

### 4.1. Study Sites and Sample Trees

The study was conducted in seven naturally regenerated stands mostly dominated by silver birch in the central part of Latvia (N 56°40′; E 25°55′), which is situated in the hemiboreal forest zone of the Eastern Baltic region (Table 2). The stands were located within a research forest massif within a range of 3 km and were growing on flat terrain under lowland conditions (<120 m a.s.l.) on mesotrophic freely draining mineral and eutrophic drained deep peat soils (according to field survey). The stand area ranged from 0.71 to 6.15 ha and the shape of the forest compartments was regular (squares).

The climate of the study area is humid continental [49], which is predetermined by the interaction between continental and oceanic air flows [50]. In the study area, the lowest and highest monthly mean air temperature occur in January–February (−4.5 °C) and July (17.6 °C), respectively. Soil freezing starts in January and lasts until March. The mean annual sum of precipitation is 698.1 mm [51]. The wind climate is strongly influenced by westerlies from the North Atlantic [52], with the mean annual wind speed of 2.5 m s^−1^ [51] and the mean maximum wind speed at the elevation of 10 m of 17.2 m s^−1^ [53].

The studied period of frozen soil conditions was January–March of 2021, during which the monthly mean minimum air temperature ranged between −7.5 and −1.3 °C. During the studied period, freeze–thaw cycle alternated; however, the length of the period with a daily mean air temperature above 0 °C did not exceed 10 days. From January 28 till February 20, the daily mean minimum air temperature was −10.5 °C and the minimum values dropped below −20 °C for 11 days. 

In the studied stands, birch was generally admixed with Norway spruce (*Picea abies* (L.) H.Karst.), Scots pine (*Pinus sylvestris* L.), and common oak (*Quercus robur* L.); however, a higher variation in species composition was found on mineral soils, as stands on peat soils were admixed by Norway spruce only (Table 2). Stands on peat soils were denser (higher stand basal area) and the diameter at breast height of canopy trees was similar to that of the stands on mineral soil. The age of the studied stands ranged between 33 and 51 years. The studied stands had been subjected to conventional sylvicultural practice for birch, which implies a rotation period of 71 years with two to three thinnings. The target stand density of pre-commercial thinning is 2000–2500 trees per hectare, while that of first and second commercial thinnings is 1000–1500 and 600–800 trees per hectare, respectively. Accordingly, the studied stands had undergone pre-commercial and first commercial thinning (manual) and their densities ranged from 900 to 1600 trees per hectare.

In total, 59 canopy birch trees were sampled during periods with frozen and non-frozen soil conditions (Table 3). Sample trees without visual signs of mechanical damage or fungal infestation were selected in accordance to the stem diameter distribution of stands. Trees located on the edges of stands and openings were not sampled in order to exclude the edge effect on the loading resistance. For uprooted trees, the depth of soil freezing was measured on the edge of soil root plate, but in the vertical cut through the soil in close proximity for others. The depth of soil–root plate was determined as the maximum length of vertical roots under the stem.

### 4.2. Static Pulling Tests

Destructive static pulling tests were performed on de-topped trees according to Krišāns et al. [13] (Appendix A). In short, the pulling was performed using a 2-stroke motor winch (1800 Capstan Cable Winch, Nordforest, Germany) and a pulley block system of two opposite Roll Double pulleys (Edelrid, Germany). The system was anchored at a 30–40 m distance (base anchor, AP2; Appendix A) and at the half height of the sample tree (upper anchor, AP1). The de-topping of sample trees was conducted 1 m above the half height (TH) in order to minimize the influence of wind-loading and tree weight on measurements of stability variables, which were performed using the TreeQinetic System (Argus Electronic GmbH, Rostock, Germany). The angle of the pulling line and pulling force were measured using a dynamometer positioned above the pulley block system. Stem deflection was recorded by inclinometers at the stem base and at 5 m height. Compression of stem was measured by a strain gauge (Elastometer) at the height of 1 m (Appendix A).

### 4.3. Data Processing and Analysis

For the evaluation of the loading resistance of trees, basal bending moment (BBM, in kNm) was estimated based on pulling force and the angle of the pulling line as follows:

BBM = F × h_AP1_ × cos(median(α_line_))
(1)


N_Δ_ = N_5m_ − N_base_
(2)


The stability proxy PF was estimated as the end of proportional increase between BBM and N_Δ_, [16,17]. The SF was considered as the maximum BBM at the fatal tree failure either by uprooting or stem breakage. 

The stem stiffness of the sample trees was characterized by the modulus of elasticity (MOE), which was calculated as follows [54]: (3)$$\mathrm{MOE}=\frac{BBM\times y}{I\times e}$$
where *BBM* is estimated for the height of strain gauge, *y* is the distance between the stem center and the measurement axis of the strain gauge, *I* is the area moment of inertia of the stem cross-section, and *e* is the strain measurement. 

Flexural strength of trees which had stem breakage was characterized by the modulus of rupture (MOR), which was estimated according to Peltola et al. [8]:(4)$$\mathrm{MOR}=\frac{32\times \left({\mathrm{BBM}}_{\mathrm{SF}}/{\mathrm{DBH}}^{3}\right)}{\pi }$$
where BBM_SF_ is the basal bending moment at secondary failure and DBH is the stem diameter at breast height.

The volume of soil–root plate (V_roots_) was approximated as an elliptical paraboloid as follows:(5)$$V=\left(\frac{1}{2}\right)\cdot \pi \cdot a\cdot b\cdot h$$
where *a* and *b* are the longest and shortest radii of the root-plate, respectively, and *h* is the depth of the soil–root plate. 

The differences in BBM_PF_ and BBM_SF_ (BBM at PF and SF, respectively), MOE, MOR, and V_roots_ between soil types and frozen/non-frozen soil conditions were evaluated using linear mixed effects models. The models in general form were:
(6)*y_ij_* = *µ* + *sf_j_* + *st_j_* + *sf_j_* × *st_j_* + (*site_j_*) + *ε_ij_*

where *sf_j_* and *st_j_* are the fixed effects of frozen/non-frozen soil conditions and soil type, respectively, and *sf_j_ × st_j_* is the interaction between frozen/non-frozen soil conditions and soil type. Site (stand) was included as a random effect (*site_j_*) to account for the imbalanced number of sampled trees and local specifics. The BBM and soil–root plate were expressed per tree size, represented by stemwood volume, which was calculated according to the local equation:

V_stem_ = 0.0000909 × H^0.71677^ × DBH^0.16692 × 0.4343 × ln(H) + 1.7570^
(7)

where H is tree height (in m) and DBH is stem diameter at breast height (in cm). 

The model was fit by the maximum likelihood approach. The significance of fixed effects was estimated using Wald’s χ^2^ test. Data were analyzed in R software (v. 4.1.0.) [55] using the packages “lme4” [56], “lmerTest” [57], and “MuMIn” [58].

## 5. Conclusions

Frozen soil conditions substantially increased the mechanical stability of silver birch on both freely draining mineral and drained deep peat soils in the Eastern Baltic region. The hypothesis of the study was rejected, as frozen soil conditions had a greater effect on the loading resistance of birch growing on mineral soils than on peat (increases of 38% and 28%, respectively). Furthermore, frozen conditions not only affected the soil–root anchorage but also the stem strength, thus amplifying the effects of winter warming on tree wind resistance. Nevertheless, Eastern Baltic silver birch showed plastic adaptations in mechanical stability to shifting soil conditions, implying their higher potential to resist fatal failure, particularly on peat soil. The estimated lower effect of freezing conditions suggest preconditions for the planning of the regeneration of stands on loose and moist soils within the Eastern Baltic region. Nonetheless, considering the shortening periods of frozen soil, proactive adaptive management aiding individual tree stability is required.

## Figures and Tables

**Figure 1 plants-11-01174-f001:**
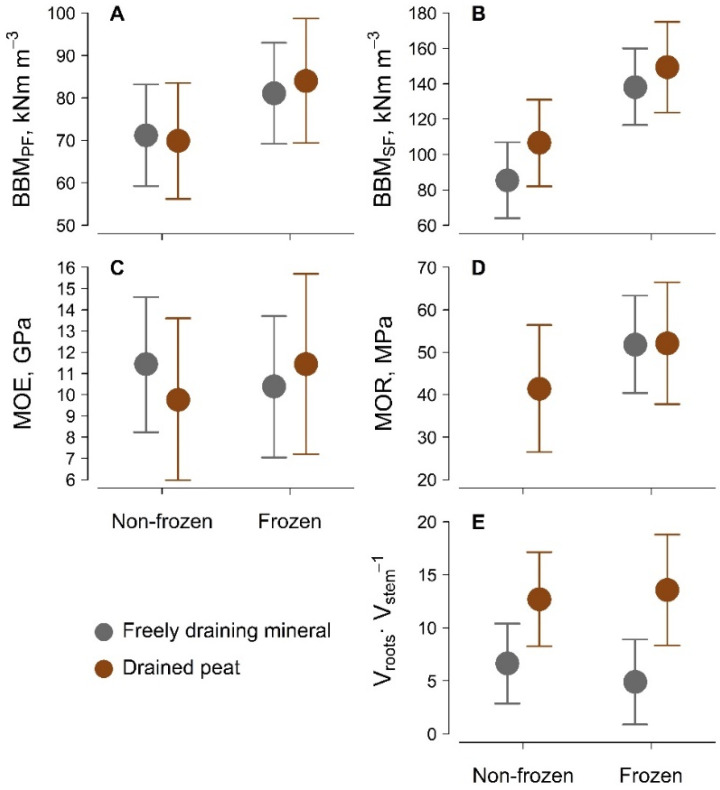
Estimated marginal mean (±95% confidence interval) basal bending moment per stemwood volume at primary (BBM_PF_) (**A**) and secondary failures (BBM_SF_) (**B**), the modulus of elasticity (MOE) (**C**), the modulus of rupture (MOR) (**D**), and the ratio of soil–root plate and stemwood volumes (**E**) according to soil type and frozen/non-frozen soil conditions for Eastern Baltic silver birch on freely draining mineral and drained peat soils. In D, one class is missing due to the insufficient sample size.

**Table 1 plants-11-01174-t001:** Strength (Wald’s χ^2^) and significance of the fixed effects (model ANOVA) of frozen/non-frozen soil conditions and soil type and their interaction on the basal bending moment at primary (BBM_PF_) and secondary failures (BBM_SF_), the modulus of elasticity (MOE) and rupture (MOR), and the volume of soil–root plate, as well as random variances of site and model performance (R^2^) for Eastern Baltic silver birch on freely draining mineral and drained peat soils. σ^2^—total variance of response; τ_00_—variance related to random effects (site); ICC—intraclass correlation coefficient.

	BBM_PF_	BBM_SF_	MOE	MOR	V_roots_
Predictors (χ^2^)					
(Intercept)	322.13 ***	155.85 ***	105.48 ***	143.90 ***	35.29 ***
Soil freezing conditions	3.85 *	45.61 ***	0.42	6.82 **	1.29
Soil type	0.04	4.00 *	0.95	-	11.10 ***
Soil freezing conditions by soil type	0.28	0.64	1.13	-	1.04
Random Effects					
σ^2^	220.67	526.70	22.32	69.45	13.91
τ_00_	16.39	75.95	0.00	10.80	1.89
ICC	0.07	0.13	0.00	0.13	0.12
n_site_	7	7	7	3	7
Observations	58	58	57	17	42
Marginal R^2^	0.13	0.53	0.02	0.28	0.43
Conditional R^2^	0.19	0.59	0.02	0.37	0.50

Significance levels: *** *p* < 0.001; ** *p* < 0.01; * *p* < 0.05.

**Table 2 plants-11-01174-t002:** Tree species composition (proportion from the total basal area), age, area, mean (±standard error) diameter at breast height (DBH), and basal area (G), as well as the number of sampled trees under frozen/non-frozen soil conditions in the studied stands with Eastern Baltic silver birch on freely draining mineral and drained peat soils. Tree species abbreviated as: B—silver birch (*Betula pendula* Roth.); S—Norway spruce (*Picea abies* (L.) H.Karst.); P—Scots pine (*Pinus sylvestris* L.); A—common aspen (*Populus tremula* L.); O—common oak (*Quercus robur* L.); OT—other species (*Alnus incana* (L.) Moench., *Sorbus aucuparia* L., *Salix caprea* L., *Crataegus rhipidophylla* Gand.).

Site No.	Species Composition (%)	Stand Age (Years)	Stand Area (ha)	DBH (cm)	G (m^2^ ha^−1^)	Frozen Soil Tree n	Non-Frozen Soil Tree n
Mineral							
1	B (33), S (38), P (26), A (3)	51	0.71	19.8 ± 0.7	14.4 ± 4.9	3	5
2	B (68), S (28), P (3), O (1)	39	0.92	19.7 ± 0.7	18.3 ± 4.4	5	5
3	B (78), S (4), O (14), OT (4)	39	1.07	19.6 ± 0.5	12.5 ± 2.9	4	5
4	B (57), S (39), P (3), OT (1)	34	6.15	19.1 ± 0.3	19.1 ± 2.7	4	4
Peat							
5	B (79), S (21)	33	0.62	18.7 ± 0.2	27.4 ± 0.9	2	5
6	B (86), S (14)	35	3.13	18.6 ± 0.1	23.3 ± 0.5	5	5
7	B (60), S (40)	51	1.50	24.0 ± 0.3	27.9 ± 1.3	3	3

**Table 3 plants-11-01174-t003:** The number (Tree n), mean (±standard error) stem diameter at breast height (DBH), height (H), stemwood volume (V_stem_), and soil–root plate volume (V_roots_) of sampled trees of Eastern Baltic silver birch on freely draining mineral and drained peat soils and soil frost depth.

Soil	Tree n	DBH (cm)	H (m)	V_stem_ (m^3^)	V_roots_ (m^3^)	Root Depth (m)	Frozen Soil Depth (cm)
Mineral							
Non-frozen	19	19.7 ± 0.6	22.5 ± 0.5	0.32 ± 0.03	2.17 ± 0.31	0.71 ± 0.04	-
Frozen	16	19.8 ± 1.0	22.9 ± 0.7	0.34 ± 0.04	1.43 ± 0.25	0.67 ± 0.03	34 ± 1
Peat							
Non-frozen	13	20.8 ± 1.1	22.7 ± 0.5	0.37 ± 0.04	4.50 ± 0.60	0.74 ± 0.02	-
Frozen	10	21.8 ± 1.5	22.5 ± 0.9	0.41 ± 0.07	5.16 ± 0.96	0.70 ± 0.04	20 ± 2

## Data Availability

Data available on request due to restrictions, e.g., privacy or ethical.

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
