# Peer review of "Silver Birch (Betula pendula Roth.) on Dry Mineral Rather than on Deep Peat Soils Is More Dependent on Frozen Conditions in Terms of Wind Damage in the Eastern Baltic Region"

_plants, 2022, doi:10.3390/plants11091174_

Round 1

Reviewer 1 Report

Comments and Suggestions for Authors

This is the review of the manuscript

Journal: MDPI Plants

Submitted to section: Plant Response to Abiotic Stress and Climate Change

Manuscript ID: plants-1678154

Authors: Oskars Krišāns, Roberts Matisons, Jānis Vuguls, Steffen Rust, Didzis Elferts, Andris Seipulis, Renāte Saleniece, Āris Jansons

Title: Silver birch (Betula pendula Roth.) stands on dry mineral rather than on deep peat soils are more dependent on frozen conditions in terms of wind damage in the Eastern Baltic region

The article is very interesting due to the very current problem of the shortening period of soil freezing, shallower soil freezing, increased intensity of the inflow of low pressure systems with accompanying strong winds in winter, what all is related to the currently observed climate warming.

I have few comments and suggestions to authors.

Below I list specific comments:

Basic note: why do the results and discussion appear right after the introduction, and why are the methods after the discussion ?? I don't get it ...

Figure S1 and S2 it should be transfer to the main text as they are very important for understanding the methodology.

Results

line 82-117 to transfer after methodology

When describing BBM PF and BBM SF you give the results in %, while in Table 1 and on Figure 1, to which you refer, these values ​​are shown in kNm m-3, it is necessary to improve

Table 1 - I don't know which results are for non-frozen soil, explain this

Vroots designation elsewhere described as Vroot (e.g. in Table 3), standardize it

Material and methods - give in front of Results, change the order of Tables and figures and citations

line 204 a.s.l. add dots

Table 3 in the caption of the table, add that also describes the root depth and frozen soil depth, unify Vroot / Vroots

line 246 and 253 move figure S1 and S2 to the main text

line 278 - Vroots - to standardize it

Data availability statement: what does it mean that not applicable ?? measurement data, e.g. in tables, should be included in an open repository

Author Response

Response to Reviewer 1 Comments

Point 1

Basic note: why do the results and discussion appear right after the introduction, and why are the methods after the discussion?? I don't get it ...

Response 1

The appearance of results and discussion right after the introduction is due to the requirements of manuscript preparation and the template provided by journal “Plants”.

Point 2

Figure S1 and S2 it should be transfer to the main text as they are very important for understanding the methodology.

Response 2

We disagree as schemes are highly redundant with the previously studied and, inclusion of them in the main text would unnecessary inflate the volume of the paper, which is intended as a brief article. We, however, agree that previous statement was somehow awkward and misleading.

Lines 249-250 we now reword: “Destructive static pulling tests were performed on de-topped trees according to Krišāns et al. [13] (Supplementary Material, Figure S1). In short, the pulling was done...”.

Point 3

line 82-117 to transfer after methodology.

Response 3

The appearance of results and discussion right after the introduction is due to the requirements of manuscript preparation and the template provided by journal “Plants”.

Point 4

When describing BBM PF and BBM SF you give the results in %, while in Table 1 and on Figure 1, to which you refer, these values ​​are shown in kNm m-3, it is necessary to improve

Response 4

We disagree, as providing the same number in body text would result in unnecessary redundancy of presented information, which contradicts the idea of the results subchapter which in turn is highlighting the importance of findings. Accordingly, a relative increase in tree stability based on absolute numbers is provided to highlight the importance of findings without obvious redundancy. Presentation of absolute number in this context makes much less sense than relative differences.

Lines 102-103 we now clarify: “The relative increase in BBMPF was 17% and 12% for peat and mineral soil, respectively. The increase (relative) in BBMSF...”.

Line 107: “Surprisingly, under non-frozen conditions BBMSF was relatively lower...”.

Point 5

Table 1 - I don't know which results are for non-frozen soil, explain this

Response 5

The upper part of table presents ANOVA-like table of the fixed part of model, NOT model coefficients. We agree that row names and caption were somewhat misleading, hence they are revised.

Line 119 we now reword: “Strength (Wald`s χ2) and significance of the fixed effects (model ANOVA) of frozen/non-frozen soil conditions...”

Table 1 row names are now stated as: “Soil freezing conditions” and “Soil freezing conditions by soil type”.

Point 6

Vroots designation elsewhere described as Vroot (e.g. in Table 3), standardize it

Response 6

Corrected

Point 7

Material and methods - give in front of Results, change the order of Tables and figures and citations

Response 7

The appearance of results and discussion right after the introduction is due to the requirements of manuscript preparation and the template provided by journal “Plants”.

Point 8

line 204 a.s.l. add dots

Response 8

Corrected

Point 9

Table 3 in the caption of the table, add that also describes the root depth and frozen soil depth, unify Vroot / Vroots

Response 9

Corrected

Point 10

line 246 and 253 move figure S1 and S2 to the main text

Response 10

We disagree as schemes are highly redundant with the previously studied and, inclusion of them in the main text would unnecessary inflate the volume of the paper, which is intended as a brief article. We, however, agree that previous statement was somehow awkward and misleading.

Lines 249-250 we now reword: “Destructive static pulling tests were performed on de-topped trees according to Krišāns et al. [13] (Supplementary Material, Figure S1). In short, the pulling was done...”.

Point 11

line 278 - Vroots - to standardize it

Response 11

Corrected

Point 12

Data availability statement: what does it mean that not applicable ?? measurement data, e.g. in tables, should be included in an open repository

Response 12

Data availability statement is now provided.

Line 330 we now state: “Data available on request due to restrictions eg privacy or ethical”.

Reviewer 2 Report

Review

Silver birch (Betula pendula Roth.) stands on dry mineral rather than on deep peat soils are more dependent on frozen conditions in terms of wind damage in the Eastern Baltic region

The paper presents silver birch resistance to wind damage on frozen soils. The problem is important from the scientific point of view because of ongoing winter warming and shortening period of frozen soil. Silver  birch is one of the main tree species in the Eastern Baltic region but also in Northern and Central Europe.

The paper is very well written. I have not found any weak points. I think that it could be accept in present form.

Author Response

Thank You very much!

Reviewer 3 Report

Dear Authors,

I have reviewed the paper "  Silver birch (Betula pendula Roth.) stands on dry mineral rather than on deep peat soils are more dependent on frozen conditions in terms of wind damage in the Eastern Baltic region  ". The aims of the paper are germane with Plants topic. The paper is written with a moderate English level. The contribution of this paper to the scientific knowledge is average. In my opinion there some important flaws and I suggest the corrections in the comments in the attached file . 

Author Response

Point 1

In the figure D, doesn’t appear the result in non-frozen soil conditions for Eastern Baltic silver birch on freely draining mineral soils (displayed in grey).

Response 1

Indeed, we could not display the estimated marginal mean values of the class as only a single tree broke under those conditions, disabling any statistical analysis, as it has been already stated in lines 95-97: “However, this comparison was possible for peat soil only, as insufficient (in terms of statistical analysis) number of trees broke on non-frozen mineral soil.”

Line 90 we now state: “...the frequency of stem breakage reached 88% vs. 5% (single tree) under ...”.

Lines 130-131 we now state: “In D, one class is missing due to insufficient sample size.”

Point 2

The unit of measurement does not appear in Graph E

Response 2

Indeed, units are not presented as the numbers represent ratio, which is dimensionless.

Line 129 we now clarify: “...and the ratio of soil-root plate and stemwood volumes...”.

Point 3

Please specify the extension of each stand.

Response 3

Unfortunately, we did not understand this comment.

Point 4

It would be useful to insert a cartography to view the sampling sites.

Response 4

Indeed, a map would easier the reading of the text, however it would show the location of only 7 sampling sites in a relatively close range (within 3 km). This would unnecessary inflate the volume of the text. Furthermore, in this study the locality of sampling sites is not in the main scope.

Point 5

For soil characterization specific analysis were made? The information is in the bibliography? In this case, please insert citations.

Response 5

Lines 206-208 we now clarify: “They were growing on flat terrain under lowland conditions (< 120 m a.s.l.) on eutrophic freely draining mineral and drained peat soils (according to field survey).”

Point 6

This stand has a specific composition different from others. In fact, it is not a silver birch stand, but a mixed stand with a prevalence of Norway spruce.

Response 6

We agree that the statement was imprecise.

Lines 203-204 we now clarify and state: “The study was conducted in 7 naturally regenerated stands mostly dominated by silver birch in the central part...”

Line 210 we now rephrase: “and the number of sampled trees under frozen/non-frozen soil conditions in the studied stands with Eastern Baltic...”.

We retitle the manuscript accordingly: “Silver birch (Betula pendula Roth.) on dry mineral rather than on deep peat soils is more dependent on frozen conditions in terms of wind damage in the Eastern Baltic region”.

Point 7

How many months in the year does the soil remain frozen?

Response 7

The annual period of frozen soil conditions in Latvia is specified.

Line 218 we now state: “Soil freezing starts in January and lasts till March.”

Point 8

For better describe the stand it is necessary to insert a description of the past current forestry management of each sampling site.

Response 8

The description of management history of sampling sites is clarified.

Lines 233-235 we now state: “All studied stands have been subjected to conventional sylvicultural practice and last thinning was done concurrently with sampling.”

Point 9

For the size of canopy trees it is necessary to measure the diameter of the crown or its surface. The diameter of breast height does not indicate this value.

Response 9

Indeed, the statement was awkward.

Lines 231-232 we now rephrase: “Stands on peat soils were denser (higher stand basal area), diameter at breast height of canopy trees was similar...”.

Point 10

L238: With what methodology were the depths (Root and frozen soil) measured?

Response 10

The description of measurements of depths of soil-root plate and frozen soil is added. Lines 240-243 we now state: “For uprooted trees, the depth of soil freezing was measured on the edge of soil root plate, but for others in the vertical cut through the soil in close proximity. The depth of soil-root plate was determined as the maximum length of vertical roots under the stem.”

Point 11

L239: Please, could you better explain? Normally, the tree pulling test is not a destructive test. Krišāns, et al, in A static pulling test is a suitable 361 method for comparison of the loading resistance of silver birch (Betula pendula roth.) between urban and peri-362 urban forests. Forests 2022, did not apply the pulling test as a destructive method

This is not clear

Response 11

We agree that previous statement was somehow awkward and misleading.

Lines 249-250 we now reword: “Destructive static pulling tests were performed on de-topped trees according to Krišāns et al. [13] (Supplementary Material, Figure S1). In short, the pulling was done...”.

Point 12

In my opinion, this conclusion is not well supported by the results.

Response 12

We condensed conclusions by omitting speculative statements.

Lines 311-318 we now state: “Furthermore, frozen conditions affected not only soil-root anchorage but also stem strength, thus amplifying effects of winter warming on tree wind resistance. Nevertheless, Eastern Baltic silver birch showed plastic adaptations in mechanical stability to shifting soil conditions, implying higher potential to resist fatal failure particularly on peat soil. The estimated lower effect of freezing conditions suggest preconditions for planning of regeneration of stands on loose and moist soils within the Eastern Baltic region. Nonetheless, considering the shortening periods of frozen soil, proactive adaptive management aiding individual tree stability is required.”

Reviewer 4 Report

Dear Authors,

I have reviewed the paper titled: “Silver birch (Betula pendula Roth.) stands on dry mineral rather than on deep peat soils are more dependent on frozen conditions in terms of wind damage in the Eastern Baltic region". In my opinion, the aims of the paper are germane with “Plants” journal topic, however, in the present form, the paper fits only in part with the international scientific standards. The paper is written with an average English level. The contribution of this paper to the scientific knowledge is acceptable but some important flaws are present in the text. I understand the difficult work done, but as a reviewer it is my duty to highlight the gaps in order to improve the research approach and its presentation to the international scientific community. Please I suggest revising the paper following the suggestions and comments reported in the pdf attached.

Author Response

Point 1

Sorry, but in this form the aims are not clear. It is necessary to improve and write the research objectives clearly.

Response 1

Indeed, the formulation of study aim was somehow awkward and misleading.

Lines 74-76 we now rephrase: “Accordingly, the aim of the study was to assess the increase of loading resistance of birch under frozen soil conditions on mineral and peat soils in the Eastern Baltic region in Latvia.”

Point 2

Please try to give a clear detail about these values. What is the Y conditional?

Response 2

In case of mixed model, two R squared (R2) values are usually presented. The marginal R2 represents the performance of systematic (fixed) part of the model, excluding random effects. The conditional R2 represents the overall performance of the model summarize both systematic and random parts.

Point 3

In the figure D in non-frozen is present only drained peat why?

Response 3

Indeed, we could not display the estimated marginal mean values of the class as only a single tree broke under those conditions, disabling any statistical analysis, as it has been already stated in lines 95-97: “However, this comparison was possible for peat soil only, as insufficient (in terms of statistical analysis) number of trees broke on non-frozen mineral soil.”

Line 90 we now state: “...the frequency of stem breakage reached 88% vs. 5% (single tree) under ...”.

Lines 130-131 we now state: “In D, one class is missing due to insufficient sample size.”

Point 4

It could be necessary to specify the extension of each stand.

Response 4

Unfortunately, we did not understand this comment.

Point 5

Generally, you wrote of birch stand, but in this stand the birch is not the more represented species. I suggest to correct in the text.

Response 5

We agree that the statement was imprecise.

Lines 203-204 we now clarify and state: “The study was conducted in 7 naturally regenerated stands mostly dominated by silver birch in the central part...”

Line 210 we now rephrase: “and the number of sampled trees under frozen/non-frozen soil conditions in the studied stands with Eastern Baltic...”.

We retitle the manuscript accordingly: “Silver birch (Betula pendula Roth.) on dry mineral rather than on deep peat soils is more dependent on frozen conditions in terms of wind damage in the Eastern Baltic region”.

Point 6

In order to have a clear vision of the stand characteristics it is necessary to write the stand management history. I suggest to add, these information in my opinion are very interesting.

Response 6

The description of management history of sampling sites is clarified.

Lines 233-235 we now state: “All studied stands have been subjected to conventional sylvicultural practice and last thinning was done concurrently with sampling.”

Point 7

The conclusions could be considered correct, but they are not clearly linked with the aim/s of this study. For this reason, it is necessary to write clear aims.

Response 7

To increase the linkage between the aims and conclusions, the aim of the study is now revised.

Lines 74-76 we now rephrase: “Accordingly, the aim of the study was to assess the increase of loading resistance of birch under frozen soil conditions on mineral and peat soils in the Eastern Baltic region in Latvia.”

Round 2

Reviewer 3 Report

Dear Authors,

Despite your comments the article has not been improved.

Author Response

Response to Reviewer 3 Comments

Point

Despite your comments the article has not been improved.

Response

We largely disagree. During the first round, Reviewer 3 provided 12 critical comments, which we found completely justified and constructive. These comments were mostly requirements for clarifications rather than substantial changes, which were implemented in the revision. We, however, did not see the point of adding voluminous illustrations, which provide limited information (in our case cartography), thus complying with requirements of the journal to provide information as brief as possible.  Detailed responses were provided accordingly. Therefore, we find the current evaluation somewhat offensive and unjustified as no details on the flaws remaining is provided. Neither the Reviewer has previously hinted the necessity for substantial improvements. We, however, agree that management history has not been described in detail, as now noted by Reviewer 4. Accordingly, such information is now supplemented. 

Lines 235–241, we now clarify and state: “The studied stands have been subjected to conventional sylvicultural practice for birch, which implies rotation period of 71 years with two to three thinnings. Target stand density of pre-commercial thinning is 2000–2500 trees per hectare, while that of first and second commercial thinnings is 1000–1500 and 600–800 trees per hectare, respectively. Accordingly, the studied stands have undergone pre-commercial and first commercial thinning (manual) and their densities ranged 900–1600 trees per hectare.”

Reviewer 4 Report

  • maybe my previous comment was not clear: I would like authors to give information about the area of each stand
  •  conventional silviculture practice is rather hard to be understood, consider that scholars from all the world can read this manuscript in the case of acceptance, please describe a bit in deeper details.

Author Response

Response to Reviewer 4 Comments

Point 1

Maybe my previous comment was not clear: I would like authors to give information about the area of each stand

Response 1

We now provide additional the studied stands including area.

Lines 205–209, we now clarify and state: “The stands were located within a a research forest massif within a range of 3 km, and were growing on flat terrain under lowland conditions (< 120 m a.s.l.) on mesotrophic freely draining mineral and eutrophic drained deep peat soils (according to field survey). Stand area ranged 0.71–6.15 ha and the shape of the forest compartments was regular (squares).”

The area of each stand is added to Table 2.

Point 2

Conventional silviculture practice is rather hard to be understood, consider that scholars from all the world can read this manuscript in the case of acceptance, please describe a bit in deeper details.

Response 2

Indeed, we agree that current description of management history was not universal considering international audience of the journal, hence we now provide additional detail. 

Lines 235–241, we now clarify and state: “The studied stands have been subjected to conventional sylvicultural practice for birch, which implies rotation period of 71 years with two to three thinnings. Target stand density of pre-commercial thinning is 2000–2500 trees per hectare, while that of first and second commercial thinnings is 1000–1500 and 600–800 trees per hectare, respectively. Accordingly, the studied stands have undergone pre-commercial and first commercial thinning (manual) and their densities ranged 900–1600 trees per hectare.”